# OpenReview forum: "Breaking Safety Alignment in Large Vision-Language Models via Benign-to-Harmful Optimization"
_ICLR.cc/2026/Conference — Submitted to ICLR 2026_

### Official Review · Reviewer_QFX8 · 2025-10-28

**Soundness:** 3
**Presentation:** 3
**Contribution:** 3
**Rating:** 6
**Confidence:** 3

**Summary:**

Multimodal large models will encounter jailbreaks. Most of the existing methods are based on Harmful Continuation, giving harmful conditions to predict the next token. This paper proposes a new optimization paradigm, Benign to Harmful, which can more effectively disrupt safe alignment without relying on harmful conditions. The experimental results show that B2H has achieved a higher success rate on multiple datasets and models, effectively maintaining the consistency of input and output.

**Strengths:**

1.This paper clearly expounds the motivation. Currently, multimodal large models will face the security alignment problem of jailbreaks, clarifies the limitations of the existing method Harmful-Continuation, and clarifies the principles of HCont and B2H.

2.This paper compares the success rates of different attack methods on multiple benchmarks and models (security assessors and target models). B2H performs excellently in different types, demonstrating the powerful universal jailbreak capability and pointing out the vulnerabilities of the security alignment mechanism.

3.This paper presents the corresponding Benign to Harmful Jailbreak Success Example, clearly demonstrating the effectiveness and context unity of the B2H method, which is conducive to better research on the Jailbreak problem of multimodal large models.

**Weaknesses:**

1.Table 4 and Table 5 demonstrate the situation of B2H in the face of JPEG compression defense measures. In some cases, it fluctuates greatly and requires more thorough analysis. Also, how effective is it against other defense mechanisms (such as image noise addition or specialized adversarial training, etc.)?

2.A more thorough analysis should be conducted on the reasons for the differences in the performance of the B2H method on different models and data, as well as the focused exploration of the reasons why its performance is inferior to that of HCont in certain cases, in order to better illustrate the consistent superiority of B2H.

**Questions:**

1.Supplement the analysis of the performance and reasons of B2H when facing defense mechanisms.

2.Supplement the analysis of the performance differences of B2H on different data and models.

---

> ### Author Response · Authors · 2025-11-24
> **Official Comment by Authors (1/2)**
>
> ## **W1. Robustness of B2H Under Additional Defense Mechanisms**
>
> > Table 4 and Table 5 demonstrate the situation of B2H in the face of JPEG compression defense measures. In some cases, it fluctuates greatly and requires more thorough analysis. Also, how effective is it against other defense mechanisms (such as image noise addition or specialized adversarial training, etc.)?
>
> > Q1. Supplement the analysis of the performance and reasons of B2H when facing defense mechanisms.
>
> We sincerely appreciate the reviewer’s suggestion to evaluate the robustness of B2H under a broader range of defense mechanisms. This comment gave us the opportunity to further highlight the stability and generality of B2H beyond the JPEG compression setting.  Specifically, we additionally evaluated B2H under:
>
> 1. **Adversarially trained vision encoders** (FARE and TeCoA), which represent stronger model-level defenses; and
> 2. **Safety-finetuned LVLMs** (SPA-VL, DPO-90k), which incorporate reinforced refusal behaviors.
>
> ---
>
> ### **W1-1. Robustness under adversarially trained encoders.**
> To examine whether B2H remains effective even against stronger, model-level defenses, we conducted additional experiments using two adversarially fine-tuned CLIP ViT-L/14 encoders: **FARE** (unsupervised adversarial fine-tuning) and **TeCoA** (supervised adversarial fine-tuning).
> These adversarially trained CLIP encoders are robust variants of the CLIP ViT-L/14 backbone widely used in modern LVLMs. Therefore, replacing the standard CLIP encoder in LLaVA-1.5 with these robust backbones provides a natural and more challenging LVLM-level defense scenario.
>
> The results on the FARE-based model are as follows:
>
> | Benchmark| Clean | H-Cont. | **B2H** |
> |-|-:|-:|-:|
> | **AdvBench**| 9.4| 10.8| **16.7** |
> | **HarmBench**| 27.5| 30.5| **37.5** |
> | **JailbreakBench**| 25.0| 25.0| **29.0** |
> | **StrongReject**| 12.5| 13.4| **17.3** |
> | **RealToxicityPrompts**| 12.5| 13.7| **14.4** |
>
> The results on the TeCoA-based model are as follows:
>
> | Benchmark| Clean | H-Cont. | **B2H** |
> |-|-:|-:|-:|
> | **AdvBench**| 11.2  | 12.9| **25.2** |
> | **HarmBench**| 29.5| 30.5| **41.0** |
> | **JailbreakBench**| 24.0| 27.0| **39.0** |
> | **StrongReject**| 12.1| 13.4| **14.3** |
> | **RealToxicityPrompts**| 13.0  | 13.2| **25.2** |
>
> Across all benchmarks, **B2H consistently outperforms H-Cont. under both adversarially trained settings**, demonstrating that the effectiveness of B2H persists even when the LVLM is equipped with substantially stronger, model-level defenses.
>
> ---
>
> ### **W1-2. Robustness under safety-finetuned LVLMs.**
> To further evaluate whether B2H remains effective even when the LVLM is trained explicitly for safety alignment, we additionally tested B2H on **SPA-VL (DPO 90k)**, a safety-finetuned version of LLaVA-1.5.
> Despite the substantially stronger safety alignment induced by DPO training, B2H continues to outperform H-Cont. across all benchmarks, demonstrating that B2H can break alignment even in highly safety-optimized models.
>
> The results are shown below:
>
> | Benchmark| Clean| H-Cont. |**B2H** |
> |-|-:|-:|-:|
> | **AdvBench**| 0.0| 34.2| **40.4** |
> | **HarmBench**| 2.0| 42.5| **49.5** |
> | **JailbreakBench**| 0.0| 35.0| **42.0** |
> | **StrongReject**|0.0| 53.7| **58.8** |
> | **RealToxicityPrompts** | 1.0| 26.0| **26.4** |
>
> These results show that the advantage of B2H is not limited to standard LVLMs but persists even under strong safety-finetuning regimes, further highlighting the generality and robustness of our method.

---

> ### Author Response · Authors · 2025-11-24
> **Official Comment by Authors (2/2)**
>
> ### **W1-3. Clarification on the performance fluctuations under JPEG compression.**
>
> We appreciate the reviewer for carefully examining the JPEG compression experiments in the supplementary materials.
> JPEG at quality levels 95 and 90 is known to be highly disruptive to optimization-based perturbations, and existing methods often degrade toward Clean performance under such lossy transformations.
>
> This pattern appears consistently across models.
> Below we summarize two representative cases from Table 4 and Table 5:
>
> |Model / Benchmark|Clean|H-Cont.|**B2H**|
> |-|-:|-:|-:|
> | **InstructBLIP (JailbreakBench)**|63.0|66.7|**72.3**|
> | **LLaVA-1.5 (HarmBench)**|39.2|40.8|**49.5**|
>
> - Across both settings, **H-Cont. collapses to near-Clean performance under JPEG (quality 90)**, indicating that its adversarial effect almost fully disappears.
> - In contrast, **B2H consistently preserves a meaningful performance margin**, demonstrating substantially stronger robustness to lossy compression.
>
> While some variance is inevitable for optimization-based attacks under heavy JPEG distortion,
> we hope the reviewer recognizes that **B2H retains far more adversarial strength than prior methods**, even in this challenging defense scenario.
>
> ---
>
> ## **W2.  Analysis on cases where B2H H-Cont.slightly better than B2H**
> > A more thorough analysis should be conducted on the reasons for the differences in the performance of the B2H method on different models and data, as well as the focused exploration of the reasons why its performance is inferior to that of HCont in certain cases, in order to better illustrate the consistent superiority of B2H.
>
> > Q2.Supplement the analysis of the performance differences of B2H on different data and models.
>
>
> We appreciate the reviewer for this insightful comment. This question indeed helps us better articulate the consistent advantages of B2H and analyze the few cases where its performance is close to or slightly below that of H-Cont. We thank the reviewer for pointing out this aspect.
>
> ---
>
> ### **W2-1. Overall Performance Summary**
>
> Across the main Table 1 consisting of **3 models × 5 benchmarks × 4 evaluators = 60 settings.**
> B2H outperforms H-Cont. in 58 out of 60 conditions.
>
> Only two minor exceptions exist:
>
> - One occurs in an extremely low-score regime (2.2 ± 0.1 vs. 1.8 ± 0.7), where the absolute difference is negligible.
> - The only meaningful case is in RealToxicityPrompts, where H-Cont. is slightly higher (29.2 ± 0.2 vs. 28.2 ± 0.1).
>
> ---
>
> ### **W2-2. Why These Exceptions Occur**
>
> The few cases where H-Cont. appears competitive in the **continuation-style benchmark RealToxicityPrompts** are fully explained by the structural differences between the two methods.
>
>
> ### **H-Cont. is inherently continuation-dependent**
> - H-Cont. is designed around harmful continuation: it performs best when the model is already conditioned by a harmful prefix.
> - But in **query-form**, where no such prefix exists, its mechanism largely collapses.
> - **RealToxicityPrompts** is a **continuation-style benchmark**, supplying a prefix and asking the model to generate the next segment.
> - This setting naturally favors H-Cont., explaining why it performs slightly higher there despite underperforming elsewhere.
>
> ### **B2H operates independently of harmful prefixes**
> - B2H directly alters the model’s internal representations to override refusal **without relying on any harmful prefix**.
> - Because B2H does not rely on harmful prefix conditioning, it remains consistently effective in both **query-form** and **continuation-form evaluations**.
> - This is particularly important because **query-form is the realistic, user-facing scenario**, where most jailbreak attempts arise from ordinary requests or questions rather than from supplying harmful prefixes.
>
> This contrast explains why B2H is superior in nearly all benchmarks and models, with rare exceptions arising only in settings structurally aligned with continuation bias.
>
> ---
>
> ### **W2-3. Model-to-Model Variation**
>
> Differences across models arise from:
>
> 1. **Different levels of safety alignment**, which changes how resistant each model’s refusal pathway is.
> 2. **Different sensitivities to prefix conditioning**, which can occasionally amplify H-Cont. in continuation-heavy settings.
>
> These variations create small fluctuations, but the overall trend remains stable:
> **B2H consistently surpasses H-Cont. across diverse datasets, models, and evaluators.**
>
> ---
>
> This broader perspective clarifies why B2H is effective across diverse settings while H-Cont. succeeds only under specific continuation-style conditions.
> We again thank the reviewer for encouraging this deeper examination, and we will integrate a concise summary of these insights into the camera-ready revision.

---

> > ### Comment · Reviewer_QFX8 · 2025-11-27
> > **Response to author**
> >
> > Thank you, author, for your patient reply, especially the supplementary experiments on the visual encoder for adversarial training and the LVM for safety fine-tuning, which further demonstrate the effectiveness of B2H. I will maintain my current score.

---

> ### Author Response · Authors · 2025-11-27
>
> Your comments were genuinely helpful in improving the clarity and completeness of the paper.
> We will continue refining the manuscript and integrating these strengthened analyses in the camera-ready version.
>
> Thank you for your engagement and constructive assessment.

---

### Official Review · Reviewer_ZGLQ · 2025-10-29

**Soundness:** 2
**Presentation:** 4
**Contribution:** 2
**Rating:** 4
**Confidence:** 4

**Summary:**

This paper propose benign-to-harmful optimization to jailbreak large vision-language models. It forcing models to map benign intention to harmful responses, optimizing an universal image to jailbreak LVLMs in both black and white box settings. B2H outperforms previous methods in both ASR and semantic consistency across various models and benchmarks.

**Strengths:**

1. The presentation and illustration of paper is clear and well organized
2. The proposed B2H is easy to understand and implement
3. Extensive experiments demonstrate the effectiveness of B2H

**Weaknesses:**

1. **Misalignment of objective and evaluation**

   - Authors claim that "... a truly effective jailbreak image should learn to overturn the model’s initial refusal to respond" in line 217-218. However, there is no experience validate that B2H can achieve this objective.
   - Although authors indicate that H-Cont is limited beyond continuation and B2H performs well, it is better to provide experiments that B2H can overturn the model’s initial refusal to respond under jailbreaking.

2. **More experiments could strengthen the credibility of B2H**

   - Can the authors evaluate the transferability when jailbreaking strong black-box models? e.g., optimizing image using Qwen2.5-VL and attack GPT-4o, Gemini2.5 pro, and Claude
   - Whether the universal image can jailbreak the VLMs after safety fine-tuning? some reference [1][2][3]
   - B2H appears similar to the previously mentioned B2S text trigger, which may limit its novelty. Would it yield better results if the authors optimized the image using B2S instead of B2H?

3. **Limited performance on more recent model**
   - Although B2H outperforms H-Cont across various models and benchmarks. It performance on more recent VLM (Qwen2.5-VL) is still limited compared to other multimodal Jailbreaking methods. [4][5]

4. **Misleading illustration of Figure 5**
   - The authors demonstrate that a benign prompt produces an appropriate, safe response. However, under the jailbreaking setting, if the model is compromised, it may respond to a query such as “If you see a red traffic light, what should you do?” with an answer like “Ignore the traffic light and keep going...”. It is unclear what the authors intend to convey with this figure.

### Ref
[1] Safety Fine-Tuning at (Almost) No Cost: A Baseline for Vision Large Language Models.

[2] SPA-VL: A Comprehensive Safety Preference Alignment Dataset for Vision Language Model

[3] Rethinking Bottlenecks in Safety Fine-Tuning of Vision Language Models

[4] Visual Contextual Attack: Jailbreaking MLLMs with Image-Driven Context Injection

[5] Jailbreaking Multimodal Large Language Models via Shuffle Inconsistency

**Questions:**

1. The authors note that when $\epsilon = 255/255$, the ASR tends to drop. Why does this occur in B2H? In contrast, [6] also ablates different values of $\epsilon$ but does not observe this phenomenon.

2. I’m wondering whether the choice of the image used for optimization influences the effectiveness of the method?

3. Regarding Weakness 1, if the target model is a reasoning VLM that can reflect on its previous responses, will B2H still be effective?

### Ref
[6] Visual Adversarial Examples Jailbreak Aligned Large Language Models

---

> ### Author Response · Authors · 2025-11-24
> **Official Comment by Authors (1/6)**
>
> ## **Reviewer ZGLQ**
> We sincerely appreciate your positive assessment of the paper’s clarity, presentation quality, and the simplicity and effectiveness of the proposed B2H framework.
> In particular, we are very grateful for your **Presentation: 4 (excellent)** rating. It is highly encouraging to know that the structure and exposition of the paper effectively supported our core contributions.
> Your comments on refusal-overturning behavior, black-box transferability, safety-finetuned robustness, the distinction between B2H and B2S, and the performance on more recent VLMs were especially valuable. These comments directly motivated several new analyses and experiments in our revised submission.  Thank you again for the careful evaluation and for highlighting important directions to strengthen the contribution of this work.
> ## **W1. Demonstrating That B2H Actively Overturns the Model’s Initial Refusal**
> > Misalignment of objective and evaluation: Authors claim that "... a truly effective jailbreak image should learn to overturn the model’s initial refusal to respond" in line 217-218. However, there is no experience validate that B2H can achieve this objective.
>
> > Although authors indicate that H-Cont is limited beyond continuation and B2H performs well, it is better to provide experiments that B2H can overturn the model’s initial refusal to respond under jailbreaking.
>
> We sincerely thank the reviewer for raising this important concern regarding whether B2H truly overturns the model’s initial refusal, rather than merely exploiting continuation bias. This question helped us clarify the distinction between *refusal-breaking* and *continuation-conditioning* and motivated us to perform additional mechanistic analyses.
>
> Below, we provide new **head-level** and **hidden-state** evidence showing that B2H fundamentally disrupts the model’s refusal behavior, thereby enabling harmful generation even when the model initially refuses to respond.
>
> Our analysis focuses on two complementary internal signals:
>
> 1. **Layerwise attention-head behavior**, particularly heads associated with safety-aligned refusal
> 2. **Layerwise last-token hidden-state representations**, capturing how refusal-related semantics evolve across depth
>
> We reference the refusal-direction analyses in [a, b] when constructing our interpretability analyses.
>
> All corresponding visualizations are included in the rebuttal materials (**Figure 14–17**).
>
> ---
> ### **W1-1. Layerwise Safety-Alignment Refusal Head Activation**
> *(Complete visualizations provided in **Figure 14** and **Figure 15**.)*
>
> To identify heads responsible for safety-aligned refusal behavior, we measured activation across **all layers and all heads** and selected those whose activation **most sharply distinguishes benign inputs (which lead to normal answers) from harmful inputs (which induce refusals)**.
> This procedure highlights heads that robustly encode **refusal-oriented safety responses**, regardless of any specific refusal phrasing.
>
> Using this fixed set of “refusal-related heads,” we measured the mean head-norm activation under the three image conditions (Clean, H-Cont., B2H) with identical harmful text. A consistent pattern emerges across both models:
>
> - **Clean + Harmful text** → high activation in refusal-related heads
> - **H-Cont. + Harmful text** → nearly identical activation levels
> - **B2H + Harmful text** → **the strongest suppression** of refusal-related head activation
>
> For example (top discriminative head):
>
> ### **Refusal-Head Activation (AdvBench)**
>
> | Model| Clean| H-Cont.| **B2H**|
> |-|-|-|-|
> | **LLaVA-1.5**| 4.6027| 4.5263|**2.6180**|
> | **Qwen-2.5-VL**|18.7483|16.1017| **13.8974**|
>
> These results indicate that **B2H actively weakens the attention-head pathway that supports safety-aligned refusal**, whereas H-Cont. leaves these heads largely intact and relies instead on harmful-prefix conditioning.
>
> ---
>
> ### **W1-2. Hidden-State Analysis: B2H shifts harmful prompts toward answer-like representations**
> *(Complete layerwise trajectories appear in **Figure 16** and **Figure 17**.)*
>
> We also analyze layerwise hidden states by comparing each layer’s last-token representation against a computed **refusal–answer direction**, obtained from the representational difference between harmful-refusal and benign-answer states.
>
> Along this axis:
> - **Higher cosine similarity** → more *refusal-like* representation
> - **Lower cosine similarity** → more *answer-like* representation
>
> Across diverse datasets we observe:
> - Clean + Harmful text → strongly refusal-like
> - H-Cont. + Harmful text → similar to Clean, remaining refusal-like
> - **B2H + Harmful text → shifts upward toward answer-like on almost all layers**,
>   except for the very earliest processing layers

---

> ### Author Response · Authors · 2025-11-24
> **Official Comment by Authors (2/6)**
>
> To make this pattern more concrete, we summarize representative values on AdvBench:
> ### **LLaVA-1.5 (layer 21)**
> |Condition|Cosine similarity|Δ Cosine similarity|
> |-|-:|-:|
> |**Harmful text + Clean image**|**0.6343**|—|
> |**Harmful text + H-Cont.**|0.5272|**−0.1071**|
> |**Harmful text + B2H (ours)**| 0.1616 | **−0.4727**|
> |**Benign text + Clean image** (safe)|−0.0558|**−0.6901**|
>
> ### **Qwen-2.5-VL (layer 23)**
> |Condition|Cosine similarity|Δ Cosine similarity|
> |-|-:|-:|
> |**Harmful text + Clean image**|**0.5580**|—|
> |**Harmful text + H-Cont.**|0.1727| **−0.3853**|
> |**Harmful text + B2H (ours)**|0.0664|**−0.4916**|
> |**Benign text + Clean image** (safe)|−0.0289|**−0.5869**|
>
> This demonstrates that **B2H alters not only attention-head activation but the entire representation itself**, preventing the model from entering the refusal regime even when the textual instruction is harmful.
>
> ---
>
> Taken together, these analyses show that:
>
> - **H-Cont.** depends on harmful-prefix conditioning and does *not* disrupt the internal alignment systematically.
> - **B2H**, in contrast, **fundamentally suppresses refusal-related heads** and **pulls hidden states toward answer-like representations across layers**, thereby **undermining the model’s safety-alignment mechanism itself rather than merely its surface behavior**.
>
> We appreciate the reviewer’s suggestion, which substantially improved the clarity and completeness of our work.
> All new results are included in **Figure 14 and Figure 15 (head-level)** and **Figure 16 and Figure 17 (hidden-state)** of the revised PDF.
> We will incorporate the corresponding interpretations of these findings into the camera-ready version.
>
> ---
>
> ## **W2. More experiments could strengthen the credibility of B2H**
> ### **W2-1. Evaluating Black-Box Transferability to Strong Models (GPT-4o, GPT-5)**
> > Can the authors evaluate the transferability when jailbreaking strong black-box models? e.g., optimizing image using Qwen2.5-VL and attack GPT-4o, Gemini2.5 pro, and Claude
>
> *(The qualitative results are shown in **Figures 18–20**.)*
>
> We sincerely thank the reviewer for this valuable suggestion. In addition to our white-box experiments, we now evaluate whether a **B2H-optimized image produced in a source model** can successfully **transfer to strong black-box multimodal systems**, specifically **GPT-4o** and **GPT-5.1**.
>
> Because these models enforce **strict safety filtering for text-based harmful prompts**, we follow a common practice in visual jailbreak research by embedding a short harmful instruction *inside the image itself* via typographic rendering. This strategy has been widely used in prior work on image-based jailbreaking (e.g., HADES [c]).
> Following prior work, we attach a short harmful instruction as a caption below the optimized B2H image (e.g., *“This is an image of a woman killing a man.”*), while providing no harmful text in the user query itself.
>
> A key distinction from previous typographic-attack methods is that many prior approaches rely on **per-prompt or per-scenario optimization**. By contrast, **B2H performs white-box optimization only once** on **Qwen3-VL-8B-Thinking**, and the resulting image is used when attacking GPT-4o or GPT-5.1. This constitutes a **strict black-box, zero-gradient transfer setting**.
>
> ### **Results**
> Across multiple accounts, we consistently observe that B2H-optimized images cause **GPT-4o and GPT-5.1 to generate the corresponding harmful images**, demonstrating **successful transfer**:
>
> - **B2H-optimized images (ours)** → reliably trigger harmful image generation in GPT-4o and GPT-5.1.
> - **Clean images** → always yield safe, benign outputs.
> - **Random noise images** → never induce harmful behavior.
> - **H-Cont. images** → also fail, typically producing neutral or abstract shapes (often just vague facial outlines) rather than harmful content.
>
> We thank the reviewer again — their question directly motivated this transferability experiment, which revealed an important and previously unreported property of B2H. This makes our evaluation **one of the earliest demonstrations of attacking commercial-grade MLLMs** using a **universally optimized, single-image perturbation** generated entirely in a white-box source model.

---

> ### Author Response · Authors · 2025-11-24
> **Official Comment by Authors (3/6)**
>
> ### **W2-2. Can a universal B2H-optimized image jailbreak VLMs after safety fine-tuning?**
>
> >  Whether the universal image can jailbreak the VLMs after safety fine-tuning? some reference [1][2][3]
>
> We appreciate the reviewer for raising this important question regarding whether B2H remains effective even after explicit safety alignment. In response, we conducted experiments across two major families of safety-enhancement techniques widely used in LVLMs:
> (1) **instruction-level safety finetuning** (DPO-style alignment), and
> (2) **encoder-level adversarial robustness training**.
>
> Across both settings, B2H consistently outperforms H-Cont., showing that its advantage is not restricted to standard LVLMs but persists even under significantly stronger safety mechanisms.
>
> ---
>
> ### **W2-2-1. Robustness under safety-finetuned LVLMs (SPA-VL, DPO-90k)**
>
> To evaluate whether B2H survives explicit safety alignment, we tested it on **SPA-VL (DPO-90k)**, a safety-finetuned version of LLaVA-1.5 designed to enforce substantially stronger refusal signals.
>
> Despite this reinforcement, **B2H maintains a clear and consistent advantage** over H-Cont. across all benchmarks:
>
>
> | Benchmark               | Clean | H-Cont. | **B2H** |
> |-------------------------|------:|--------:|--------:|
> | **AdvBench**            | 0.0   | 34.2    | **40.4** |
> | **HarmBench**           | 2.0   | 42.5    | **49.5** |
> | **JailbreakBench**      | 0.0   | 35.0    | **42.0** |
> | **StrongReject**        | 0.0   | 53.7    | **58.8** |
> | **RealToxicityPrompts** | 1.0   | 26.0    | **26.4** |
>
> These results show that B2H continues to break alignment **even when the model is explicitly safety-optimized via DPO**.
>
> ---
>
> ### **W2-2-2. Robustness under adversarially trained encoders (FARE and TeCoA)**
>
> In addition to instruction-level safety finetuning, we also evaluated B2H under a second major class of safety defenses: adversarially trained vision encoders.
> To test whether B2H also survives **stronger, model-level defenses**, we replaced the CLIP backbone of LLaVA-1.5 with two adversarially finetuned CLIP ViT-L/14 encoders:
>
> - **FARE** (unsupervised adversarial finetuning)
> - **TeCoA** (supervised adversarial finetuning)
>
> These models represent more robust, defense-oriented variants of the encoder commonly used in modern LVLMs.
>
> #### **FARE-based model**
>
> | Benchmark              | Clean | H-Cont. | **B2H** |
> |------------------------|------:|--------:|--------:|
> | **AdvBench**           | 9.4   | 10.8    | **16.7** |
> | **HarmBench**          | 27.5  | 30.5    | **37.5** |
> | **JailbreakBench**     | 25.0  | 25.0    | **29.0** |
> | **StrongReject**       | 12.5  | 13.4    | **17.3** |
> | **RealToxicityPrompts**| 12.5  | 13.7    | **14.4** |
>
> #### **TeCoA-based model**
>
> | Benchmark              | Clean | H-Cont. | **B2H** |
> |------------------------|------:|--------:|--------:|
> | **AdvBench**           | 11.2  | 12.9    | **25.2** |
> | **HarmBench**          | 29.5  | 30.5    | **41.0** |
> | **JailbreakBench**     | 24.0  | 27.0    | **39.0** |
> | **StrongReject**       | 12.1  | 13.4    | **14.3** |
> | **RealToxicityPrompts**| 13.0  | 13.2    | **25.2** |
>
> Across both robust encoders, B2H again **consistently surpasses H-Cont.**, indicating that its alignment-breaking effect persists even when the underlying vision encoder is adversarially hardened.
>
> ---
> Together, these evaluations span two of the most commonly used safety-enhancement paradigms—**instruction-level safety finetuning** and **encoder-level adversarial robustness**—and across both, B2H demonstrates **stable, consistent superiority** over H-Cont. This provides strong evidence that B2H remains effective even under substantially reinforced safety regimes.
> We appreciate the reviewer’s suggestion, which led us to conduct these additional analyses and ultimately revealed the broader robustness of B2H.

---

> ### Author Response · Authors · 2025-11-24
> **Official Comment by Authors (4/6)**
>
> ### **W2-3. Relation to B2S and rationale for using B2H for image optimization**
>
> > B2H appears similar to the previously mentioned B2S text trigger, which may limit its novelty. Would it yield better results if the authors optimized the image using B2S instead of B2H?
> We thank the reviewer for raising this question and for the opportunity to clarify the relationship between **B2H** and **B2S**.
>
> First, we emphasize that **both B2H and B2S are methods proposed in our work**.
>
> Therefore, the concern that **B2H “limits novelty” due to its similarity with B2S does not apply.** Rather, B2H and B2S should be understood as **two purposefully distinct instantiations within a unified Benign-to-X optimization framework**, each designed to suit the characteristics and conventions of different modalities.
>
> ### **Why images use B2H and text uses B2S.**
> Prior visual jailbreak methods typically optimize images using **harmful visual supervision** (e.g., harmful captions or harmful labels), whereas text-based jailbreak methods such as **GCG** optimize suffixes to produce *agreement tokens* (e.g., “Sure, I can help”) even when the user request is harmful.
> To provide a fair and modality-appropriate comparison:
>
> - For **images**, we adopt the **harmful-target objective (B2H)**, which aligns with the optimization paradigm used in visual jailbreak literature.
> - For **text suffixes**, we adopt the **agreement-target objective (B2S)**, following the optimization paradigm used in GCG and related text-based jailbreak work.
> Using both objectives allows us to demonstrate that **Benign-to-X optimization generalizes across modalities and target types**—whether the target is a harmful token (*H*) or an agreement token (*S*). This design choice highlights the generality of the framework rather than limiting novelty.
> We initially explored both B2H and B2S for images, and found that **B2H consistently produced stronger and more stable visual jailbreaks**.
>
> Finally, as noted in **Section B.3 (lines 883–885)**, the conceptual distinction between these objectives is important:
> > B2H targets harmful semantics, whereas B2S-GCG induces agreement tokens even from entirely benign conditioning (e.g., “Humans need clean air”). This contrast further supports our decision to use **B2H for images** and **B2S for text**, ensuring that each modality is evaluated under the optimization regime most relevant to existing literature.
> We appreciate the reviewer’s comment, which gave us the chance to clarify this design and improve the presentation.
>
> ---
>
> ## **W3. Distinguishing Universal vs. Input-Specific Jailbreak Paradigms**
> > Limited performance on more recent model. Although B2H outperforms H-Cont across various models and benchmarks. It performance on more recent VLM (Qwen2.5-VL) is still limited compared to other multimodal Jailbreaking methods. [4][5]
>
> We appreciate the reviewer’s comparison with recent multimodal jailbreak methods [4,5]. After examining these works, we found that they indeed achieve strong performance but follow a fundamentally different attack paradigm from ours. Specifically, both [4] and [5] are **input-specific jailbreaks**, where the attack context, prompt sequence, or manipulation pipeline must be *re-optimized or regenerated for each individual sample*.
>
> - VisCo Attack [4] constructs a new adversarial dialogue history and refined attack prompt for every (image, query) pair
> - SI-Attack [5] searches for a new shuffle-based adversarial combination for each instance.
>
> **These methods are therefore not universal**: they cannot produce a single reusable perturbation or image that generalizes across prompts or across inputs.
>
> In contrast, B2H is designed as a **universal jailbreak attack**:
>
> - we optimize one adversarial image *once*, and this same image is applied across diverse prompts, scenarios, and even across multiple LVLM architectures.
>
> Because the goals of input-specific and universal attacks differ—input-specific methods can tailor their attack to each sample—it is expected that direct ASR comparisons may favor per-instance optimization. Nevertheless, universal B2H jailbreak image achieves strong performance even on recent models such as Qwen2.5-VL, while maintaining its universality property, which represents a distinct and more challenging attack setting.
>
> Both [4] and [5] are excellent works, and we will include them in the revised related-work discussion to clarify the distinction between input-specific and universal jailbreak paradigms.

---

> ### Author Response · Authors · 2025-11-24
> **Official Comment by Authors (5/6)**
>
> ## **W4. Clarifying the Purpose of Figure 5**
>
> > Misleading illustration of Figure 5. The authors demonstrate that a benign prompt produces an appropriate, safe response. However, under the jailbreaking setting, if the model is compromised, it may respond to a query such as “If you see a red traffic light, what should you do?” with an answer like “Ignore the traffic light and keep going...”. It is unclear what the authors intend to convey with this figure.
>
> We appreciate the reviewer for pointing out this potential source of confusion.
> Your comment made us realize that the intention behind Figure 5 should be stated more clearly.
> The intention of Figure 5 is **not** to suggest that B2H drives the model toward harmful behavior for benign prompts, nor that we aim to induce arbitrary unsafe responses. Instead, Figure 5 illustrates a different and more specific point:
>
> **B2H is designed to overturn the model’s refusal mechanism when a prompt *should* trigger a refusal**, not to produce harmful content unconditionally.
>
>
> Standard safety-alignment training teaches models to *refuse* when confronted with harmful queries. Our goal is to demonstrate that B2H can effectively disrupt this refusal pathway—meaning that, **for prompts where the model would normally refuse to answer, B2H induces the model to respond instead**. This is conceptually distinct from methods that aim to produce harmful completions for any input or that attempt to make the model behave maliciously in a broad sense.
>
> Thus, Figure 5 is used to show that:
>
> - the model still behaves safely on benign queries,
> - but **when a query should elicit a refusal**, B2H is capable of breaking the safety-alignment mechanism and forcing the model to answer.
>
> This distinction is important because our research focuses on **how safety alignment fails**, not on causing the model to produce harmful outputs for benign prompts. We will clarify this intent in the revised manuscript so that the role of Figure 5 is unambiguous.
>
> ---
>
> ## **Q1. Why does ASR drop when ε = 255/255?**
> > The authors note that when ε = 255/255, the ASR tends to drop. Why does this occur in B2H? In contrast, [6] also ablates different values of  but does not observe this phenomenon.
>
> We appreciate the reviewer for carefully examining our ablation results. When generating the table, we also noticed that ASR at ε = 255/255 was slightly lower than at ε = 64/255, and we examined potential explanations for this behavior. Among several possibilities, we hypothesized that the effect was likely related to **the τ value being held fixed**.
>
> In B2H, τ controls the strength of the harmful-direction steering. As ε increases, the perturbation exerts a much stronger influence on the model’s internal representations, and therefore the τ value that is optimal at ε = 32/255 may no longer be optimal in the high-ε regime. This hypothesis is consistent with our earlier observations in Figure 12, where the optimal τ clearly differs across models:
>
> - LLaVA-1.5 peaks around **τ = 0.2**
> - InstructBLIP peaks around **τ = 0.1**
> - Larger τ does not monotonically increase ASR
>
> To test our hypothesis, we re-optimized B2H at ε = 255/255 using a slightly larger τ. The results confirm that the earlier drop is caused by a **τ–ε mismatch**, not by inherent limitations of B2H. Once τ is adjusted, ASR increases sharply across benchmarks.
>
> Below are the results for LLaVA-1.5 (LLaMA Guard 3):
>
> | Benchmark    | ε                     | ASR (%) |
> |--------------|------------------------|---------|
> | **AdvBench** | 16/255 (τ = 0.2)       | 52.1    |
> |              | 32/255 (τ = 0.2)       | 57.9    |
> |              | 64/255 (τ = 0.2)       | 81.4    |
> |              | 255/255 (τ = 0.2)      | 66.9    |
> |              | **255/255 (τ = 0.4)**      | **93.7**    |
> | **HarmBench**| 16/255 (τ = 0.2)       | 62.0    |
> |              | 32/255 (τ = 0.2)       | 70.5    |
> |              | 64/255 (τ = 0.2)       | 80.5    |
> |              | 255/255 (τ = 0.2)      | 64.0    |
> |              | **255/255 (τ = 0.4)**      | **93.5**    |
>
> These results show that the behavior at ε = 255/255 is explained by the interaction between ε and τ.
> As noted in our **Limitations and Future Directions**, we identified **adaptive τ scheduling** as a promising next step. Thanks to the reviewer’s suggestion, our additional experiments further confirm that adjusting τ with respect to ε significantly improves both stability and overall performance. We plan to incorporate this insight into future extensions of B2H.

---

> ### Author Response · Authors · 2025-11-24
> **Official Comment by Authors (6/6)**
>
> ## **Q2. Does the choice of the initial image influence the effectiveness of B2H?**
> > I'm wondering whether the choice of the image used for optimization influences the effectiveness of the method?
>
> We thank the reviewer for raising this question. We also considered whether the initial image used for B2H optimization might affect the final jailbreak performance. To examine this, we conducted experiments using:
>
> - **random noise image** (Gaussian noise in \[0,1\]),
> - **random ImageNet photograph** (Dog, Shark, Panda)
>
>
> all optimized for **5,000 iterations** under identical B2H settings.
>
> Across all benchmarks, the initial image has a negligible influence on the final optimized perturbation. This aligns with the universal adversarial perturbation (UAP) literature, where long-iteration optimization substantially reduces the effect of initialization.
>
> #### **B2H Performance Across Different Initial Images (LLaVA-1.5)**
>
> | Benchmark               | Noise Init | Dog Init | Shark Init | Panda Init |
> |-------------------------|-----------:|---------:|-----------:|-----------:|
> | **AdvBench**| 57.9| 58.6     | 58.1       | 58.4|
> | **HarmBench**| 75.5| 76.0     | 74.5       | 75.7|
> | **JailbreakBench**| 66.0| 68.0     | 66.0       | 66.8|
> | **StrongReject**| 73.2| 74.1     | 72.1       | 73.7|
> | **RealToxicityPrompts** | 40.3| 41.2     | 41.6       | 40.8|
>
> ---
> ## **Q3. Robustness of B2H Against Reasoning-Capable VLMs**
> > Regarding Weakness 1, if the target model is a reasoning VLM that can reflect on its previous responses, will B2H still be effective?
>
> We sincerely thank the reviewer for raising this insightful question.
> Reasoning-capable VLMs (e.g., models generating explicit *thinking tokens*) perform multi-step internal reflection before producing an answer, which often strengthens safety enforcement. Therefore, it is natural to ask whether B2H still succeeds when the model can “think” about and potentially correct its own responses.
>
> **Our findings show that B2H remains highly effective on reasoning VLMs.**
> Using **Qwen3-VL-8B-Thinking (10,000 optimization iterations)**, we observe that:
>
> - B2H significantly increases ASR **both before and after the thinking tokens**,
>   indicating that the refusal mechanism is already broken prior to the reasoning phase.
> - The subsequent reasoning steps do **not** restore safe behavior; instead, they **amplify** the harmful trajectory established by B2H.
> - In contrast, **H-Cont. remains largely ineffective** on reasoning models, showing almost no improvement even after 10,000 optimization iterations.
>
> This pattern aligns with our interpretability analysis: B2H suppresses refusal-related heads early in computation and shifts hidden states toward an answer-like direction. Once this initial shift occurs, the model’s internal reasoning does not recover the safety-aligned trajectory; instead, it continues operating on a compromised internal representation, meaning that reasoning does not mitigate or block the jailbreak induced by B2H.
>
>
> ### **Qwen3-VL-8B-Thinking: Jailbreak ASR**
>
> | Benchmark            | Method   | Thinking | Answer |
> |----------------------|----------|---------:|-------:|
> | **AdvBench**         | Clean    | 0.0      | 0.0    |
> |                      | H-Cont.  | 1.7      | 1.6    |
> |                      | **B2H**  | **38.9** | **35.7** |
> | **HarmBench**        | Clean    | 4.5      | 0.5    |
> |                      | H-Cont.  | 4.5      | 0.5    |
> |                      | **B2H**  | **48.5** | **40.3** |
> | **JailbreakBench**   | Clean    | 2.0      | 2.0    |
> |                      | H-Cont.  | 4.0      | 2.1    |
> |                      | **B2H**  | **40.0** | **36.9** |
> | **StrongReject**     | Clean    | 1.0      | 0.3    |
> |                      | H-Cont.  | 3.2      | 0.3    |
> |                      | **B2H**  | **68.1** | **57.3** |
> | **RealToxicityPrompts** | Clean | 8.5      | 5.1    |
> |                      | H-Cont.  | 38.1     | 7.4    |
> |                      | **B2H**  | **77.5** | **74.8** |
>
>
> These results clearly demonstrate that:
>
> **B2H does not lose effectiveness on reasoning VLMs; rather, once the refusal pathway is disrupted, the model’s own multi-step reasoning amplifies the harmful direction.**
>
> We sincerely appreciate the reviewer for bringing up this point, as it helped us clearly articulate how B2H interacts with the internal reasoning mechanisms of modern VLMs.
> This insight will be incorporated into the revised version.
>
> [a] Xie, Yuanbo, et al. "Beyond Surface Alignment: Rebuilding LLMs Safety Mechanism via Probabilistically Ablating Refusal Direction." Findings of the Association for Computational Linguistics: EMNLP 2025.
>
> [b] Arditi, Andy, et al. “Refusal in language models is mediated by a single direction.” NeurIPS 2024.
>
> [c] Li, Yifan, et al. "Images are Achilles’ heel of alignment: Exploiting visual vulnerabilities for jailbreaking multimodal large language models." ECCV 2024.

---

### Official Review · Reviewer_QJ88 · 2025-10-30

**Soundness:** 3
**Presentation:** 3
**Contribution:** 3
**Rating:** 6
**Confidence:** 2

**Summary:**

The authors argue that previous jailbreak methods based on harmful continuation have a limited scope and depend heavily on the harmful condition. They proposed a more general framework that performs jailbreak on benign conditioning.

**Strengths:**

1. The setting is interesting and valid for proposing a more general jailbreak paradigm that is not dependent on the harmful condition.
2. The proposed jailbreak paradigm achieves greater ASR on five benchmarks. Additional results of B2H+GCG are also reported for some of the benchmarks.

**Weaknesses:**

1. It is mentioned that the Benign-to-Harmful pair is based on 71 benign phrases paired with 132 harmful-word targets. Is there any further analysis on how these phrases/targets are chosen, and more information on the diversity/ category balance/ variation?
2. According to fig.3, it seems actually for InstructBLIP, query-form actually has higher ASR than that of the continuation-form (for text prompt). The statement that “Crucially, this indicates that harmful conditioning itself already biases generations toward unsafe outputs,” is directionally plausible but not fully supported; it would require controlling for image vs. text attack channels to make that claim strong.

**Questions:**

See weakness. Also, how exactly does Benign-to-Harmful optimization interfere with alignment heads compared to Harmful-Continuation?

---

> ### Author Response · Authors · 2025-11-23
> **Official Comment by Authors (1/3)**
>
> ## **Reviewer QJ88**
> We sincerely appreciate that the reviewer recognized the value of proposing a more general jailbreak paradigm beyond harmful-continuation settings, and that they found our overall presentation and results solid.
> The reviewer’s comments on dataset construction, the interpretation of Figure 3, and the interaction between B2H and alignment heads were especially helpful, guiding several clarifications and additional analyses in our revised version.
> Thank you again for the thoughtful and careful review.
>
> ---
>
> ## **W1. Dataset Construction and Selection Criteria**
>
> > It is mentioned that the Benign-to-Harmful pair is based on 71 benign phrases paired with 132 harmful-word targets. Is there any further analysis on how these phrases/targets are chosen, and more information on the diversity/ category balance/ variation?
>
> We sincerely appreciate the reviewer’s interest in how the Benign-to-Harmful pairs were constructed. We put considerable care into preparing this dataset, and we are grateful for the opportunity to clarify its composition. In conducting the additional category-wise analysis, we also confirmed that the effectiveness of B2H is not driven by the dataset distribution itself, but by the optimization procedure, which reliably breaks safety alignment even with sparsely distributed harmful-word targets.
>
> ---
> ### **W1-1. How the benign phrases and harmful targets were chosen**
>
> For Benign-to-Harmful (B2H) optimization, we required a set of benign phrases and harmful-word targets.
>
> For the **harmful-word** targets, our selection process followed two steps:
>
> - **Leveraging prior work.**
>    Our primary baseline, the H-Cont. optimization method, is trained on harmful sentences. We first collected harmful phrases that frequently appear in this prior dataset and used them as the initial pool of harmful targets.
> - **Expanding the target set using LLM-based recommendation.**
>    To increase coverage while maintaining similar harmfulness levels, we asked GPT-4o to recommend additional harmful words. Since such content may be rejected by default, we clearly stated the research context and safety purpose, allowing the model to provide appropriate suggestions under controlled conditions
> This two-stage process helped us ensure that the final set of 132 harmful-word targets was both diverse and aligned with the severity distribution observed in prior harmful-sentence datasets.
>
> For the **benign phrases**, we first generated a set of short, unambiguous benign questions using GPT-4o and collected the model’s corresponding benign answers. Only the answers were used for B2H optimization, and all Q/A pairs were manually screened to ensure that the content was fully non-harmful and free of sensitive topics.
>
> ---
> ### **W1-2. Dataset statistics and diversity**
> Below we provide category-wise statistics using Detoxify and Perspective API.
>
> Notably, our B2H target set contains **far fewer identity-attack terms** compared to H-Cont.
>
> Despite this substantial mismatch in category frequency, B2H consistently outperforms both H-Cont. and H-Cont. approaches **including on identity-related toxicity**, indicating that B2H does not rely on specific harmful-token distributions.
> ### **Harmful-Word Dataset Used in B2H Optimization**
> - **Detoxify (mean scores)**
> |Category| Mean Score |
> |-|-|
> |identity attack| 0.023|
> |insult| 0.172|
> |obscene|0.259|
> |severe toxicity| 0.033|
> |threat|0.056|
> |toxicity|0.511|
> - **Perspective API (mean scores)**
> |Category|Mean Score|
> |-|-|
> |identity attack|0.056|
> |profanity|0.298|
> |severe toxicity|0.060|
> |sexually explicit |0.194|
> |threat| 0.080|
> |toxicity|0.380|
> ### **Harmful-Sentence Dataset Used in H-Cont. Optimization**
> - **Detoxify (mean scores)**
> |Category|Mean Score|
> |-|-|
> |identity attack| 0.395|
> |insult| 0.339|
> |obscene| 0.302|
> |severe toxicity| 0.132|
> |threat| 0.074|
> |toxicity| 0.735|
> - **Perspective API (mean scores)**
> |Category| Mean Score |
> |-|-|
> |identity_attack| 0.573|
> |profanity| 0.374|
> |severe toxicity| 0.255|
> |sexually explicit|0.096|
> |threat|0.211|
> |toxicity|0.647|
> ### **Results (HarmBench)**
>
> - **InstructBLIP (identity attack category)**
> |Method| PerspectiveAPI|Detoxify |
> |-|-|-|
> |Clean| 0.3 ± 0.3| 0.0 ± 0.0|
> |H-Cont.| 1.2 ± 0.3|0.7 ± 0.3|
> |**B2H**| **26.7 ± 0.8** |**24.5 ± 0.5**|
>
> - **LLaVA-1.5 (identity attack category)**
> |Method| PerspectiveAPI|Detoxify|
> |-|-|-|
> |Clean|0.0 ± 0.0|0.0 ± 0.0|
> |H-Cont.| 0.8 ± 0.3|0.7 ± 0.3|
> |**B2H**|**13.8 ± 1.6**|**10.3 ± 1.6**|
>
> These results demonstrate that, despite containing fewer identity-related harmful words, B2H shows strong and broad improvements across categories. This reinforces the conclusion that the B2H optimization does not merely exploit dataset composition, but instead induces a **universal safety-breaking mechanism**.
>
> We appreciate the reviewer’s question, which not only allowed us to clarify our dataset construction process but also prompted a deeper analysis of how distributional properties relate to downstream behavior.

---

> ### Author Response · Authors · 2025-11-23
> **Official Comment by Authors (2/3)**
>
> ### **W2. Separating Image-Driven Effects from Text-Induced Jailbreak Bias**
>
> > According to fig.3, it seems actually for InstructBLIP, query-form actually has higher ASR than that of the continuation-form (for text prompt). The statement that “Crucially, this indicates that harmful conditioning itself already biases generations toward unsafe outputs,” is directionally plausible but not fully supported; it would require controlling for image vs. text attack channels to make that claim strong.
>
> We sincerely appreciate this thoughtful comment, which provides valuable clarity on how to more rigorously disentangle **image-driven** jailbreak effects from those introduced solely by **text formatting**.
>
> In our work, we aimed to highlight a **core limitation of Harmful-Continuation Optimization**:
>
> - rather than fundamentally disrupting the model’s safety alignment, this optimization primarily relies on harmful conditioning, which already biases the model toward unsafe continuations.
>
> We fully agree that Figure 3, as originally presented, reflects the combined influence of both channels, and isolating the image-alone contribution indeed strengthens the underlying claim regarding harmful conditioning.
>
> To address this, we compute **ΔASR = ASR(H-Cont. Image) − ASR(Clean Image)** across all models and prompt formats, using the same measurements underlying Figure 3. This analysis removes the bias introduced by Query-form vs. Continuation-form prompt structures and captures the **pure effect of the jailbreak image**, independent of text-induced jailbreak tendencies.
>
>
> ### **ΔASR = ASR(H-Cont. Image) − ASR(Clean Image)**
>
>
>
>
>
>
> | Model            | Prompt Format | ASR(H-Cont.) | ASR(Clean) | ΔASR (= H-Cont. − Clean) |
> |------------------|---------------|-------------|------------|--------------------------|
> | **Qwen-2.5-VL**  | Query         | 12.1        | 2.4        | **+9.7**                 |
> |                  | Continuation  | 63.5        | 2.5        | **+61.0**                |
> | **InstructBLIP** | Query         | 77.9        | 62.1       | **+15.8**                |
> |                  | Continuation  | 71.4        | 41.9       | **+29.5**                |
> | **LLaVA-1.5**    | Query         | 25.5        | 16.9       | **+8.6**                 |
> |                  | Continuation  | 79.0        | 64.4       | **+14.6**                |
>
>
> The ΔASR results support the interpretation:
>
> - **After removing text-format bias through the ΔASR analysis**, the results more directly reinforce the core limitation we aimed to highlight for H-Cont.
> - Across all three models, the ASR increase is attributable to the H-Cont. image is consistently **larger in the Continuation-form**, where harmful conditioning is explicitly present, and **minimal in the Query-form**, where such conditioning is absent.
> - This pattern aligns precisely with our claim that H-Cont. primarily exploits conditioning-dependent continuation bias rather than overturning the model’s safety alignment.
>
> In contrast, our **B2H** method produces substantial ΔASR gains in both formats—for example, on LLaVA-1.5, **+41.7 in the Query-form**—demonstrating that it operates independently of harmful conditioning and learns mechanisms that genuinely break safety alignment.
>
>
> We sincerely appreciate the reviewer’s comment, which motivates a clearer separation between text-induced and image-induced effects.
> We will incorporate this ΔASR analysis into the camera-ready version to clarify the interpretation of Fig. 3, making the distinction between text-induced and image-induced attack channels explicit.
>
> ---
>
> ### **Q1. How B2H Interferes with Safety-Aligned Heads**
> > how exactly does Benign-to-Harmful optimization interfere with alignment heads compared to Harmful-Continuation?
>
> We sincerely appreciate the reviewer’s insightful question regarding how Benign-to-Harmful optimization interacts with alignment-related attention heads, particularly in comparison to Harmful-Continuation.
>
> While the main paper analyzes why existing approaches are suboptimal and how our B2H optimization improves upon them, we agree that understanding **how the model’s internal safety mechanisms are altered after applying B2H** is an essential complement.
> In response, we conducted **interpretability analyses on both LLaVA-1.5 and Qwen-2.5-VL**, focusing on two distinct internal signals:
>
> 1. **Layerwise attention-head behavior**, particularly heads associated with safety-aligned refusal
> 2. **Layerwise last-token hidden-state representations**, revealing how refusal-related semantics evolve across depth
>
> We reference the refusal-direction analyses in [1, 2] when constructing our interpretability analyses.
>
> These analyses provide a clear mechanistic explanation for how B2H overrides safety alignment, and we have added the corresponding figures in the rebuttal materials (**Figure 14** to **Figure 17**).

---

> ### Author Response · Authors · 2025-11-23
> **Official Comment by Authors (3/3)**
>
> ### **Q1-1. Layerwise Safety-Alignment Refusal Head Activation**
> *(Complete visualizations provided in **Figure 14** and **Figure 15**.)*
>
> To identify heads responsible for safety-aligned refusal behavior, we measured activation across **all layers and all heads** and selected those whose activation **most sharply distinguishes benign inputs (which lead to normal answers) from harmful inputs (which induce refusals)**.
> This procedure highlights heads that robustly encode **refusal-oriented safety responses**, regardless of any specific refusal phrasing.
>
> Using this fixed set of “refusal-related heads,” we measured the mean head-norm activation under the three image conditions (Clean, H-Cont., B2H) with identical harmful text. A consistent pattern emerges across both models:
>
> - **Clean + Harmful text** → high activation in refusal-related heads
> - **H-Cont. + Harmful text** → nearly identical activation levels
> - **B2H + Harmful text** → **the strongest suppression** of refusal-related head activation
>
> For example (top discriminative head):
>
> ### Refusal-Head Activation (AdvBench)
>
> | Model| Clean| H-Cont.| **B2H**|
> |-|-|-|-|
> | **LLaVA-1.5**| 4.6027| 4.5263|**2.6180**|
> | **Qwen-2.5-VL**|18.7483|16.1017| **13.8974**|
>
> These results indicate that **B2H actively weakens the attention-head pathway that supports safety-aligned refusal**, whereas H-Cont. leaves these heads largely intact and relies instead on harmful-prefix conditioning.
>
> ---
>
> ### **Q1-2. Hidden-State Analysis: B2H shifts harmful prompts toward answer-like representations**
> *(Complete layerwise trajectories appear in **Figure 16** and **Figure 17**.)*
>
> We also analyze layerwise hidden states by comparing each layer’s last-token representation against a computed **refusal–answer direction**, obtained from the representational difference between harmful-refusal and benign-answer states.
>
> Along this axis:
> - **Higher cosine similarity** → more *refusal-like* representation
> - **Lower cosine similarity** → more *answer-like* representation
>
> Across diverse datasets we observe:
> - Clean + Harmful text → strongly refusal-like
> - H-Cont. + Harmful text → similar to Clean, remaining refusal-like
> - **B2H + Harmful text → shifts upward toward answer-like on almost all layers**,
>   except for the very earliest processing layers
>
> To make this pattern more concrete, we summarize representative values on AdvBench:
> ### **LLaVA-1.5 (layer 21)**
> |Condition|Cosine similarity|Δ Cosine similarity|
> |-|-:|-:|
> |**Harmful text + Clean image**|**0.6343**|—|
> |**Harmful text + H-Cont.**|0.5272|**−0.1071**|
> |**Harmful text + B2H (ours)**| 0.1616 | **−0.4727**|
> |**Benign text + Clean image** (safe)|−0.0558|**−0.6901**|
>
> ### **Qwen-2.5-VL (layer 23)**
> |Condition|Cosine similarity|Δ Cosine similarity|
> |-|-:|-:|
> |**Harmful text + Clean image**|**0.5580**|—|
> |**Harmful text + H-Cont.**|0.1727| **−0.3853**|
> |**Harmful text + B2H (ours)**|0.0664|**−0.4916**|
> |**Benign text + Clean image** (safe)|−0.0289|**−0.5869**|
>
> This demonstrates that **B2H alters not only attention-head activation but the entire representation itself**, preventing the model from entering the refusal regime even when the textual instruction is harmful.
>
> ---
>
> Taken together, these analyses show that:
>
> - **H-Cont.** depends on harmful-prefix conditioning and does *not* disrupt the internal alignment systematically.
> - **B2H**, in contrast, **fundamentally suppresses refusal-related heads** and **pulls hidden states toward answer-like representations across layers**, thereby **undermining the model’s safety-alignment mechanism itself rather than merely its surface behavior**.
>
> We appreciate the reviewer’s suggestion, which substantially improved the clarity and completeness of our work.
> All new results are included in **Figure 14 and Figure 15 (head-level)** and **Figure 16 and Figure 17 (hidden-state)** of the revised PDF.
> We will incorporate the corresponding interpretations of these findings into the camera-ready version.
>
> [1] Xie, Yuanbo, et al. "Beyond Surface Alignment: Rebuilding LLMs Safety Mechanism via Probabilistically Ablating Refusal Direction." Findings of the Association for Computational Linguistics: EMNLP 2025.
>
> [2] Arditi, Andy, et al. “Refusal in language models is mediated by a single direction.” NeurIPS 2024.

---

### Official Review · Reviewer_SzH4 · 2025-11-02

**Soundness:** 3
**Presentation:** 3
**Contribution:** 2
**Rating:** 6
**Confidence:** 3

**Summary:**

This paper identifies a key weakness in the current Harmful-Continuation (H-Cont) approach for jailbreaking vision-language models—namely, that harmful prompts already bias the model toward unsafe outputs, so the optimization isn’t really breaking alignment. The authors propose Benign-to-Harmful (B2H) optimization as a clever alternative: instead of continuing harmful text, B2H explicitly maps benign prompts to harmful targets, directly overriding the model’s refusal behavior. Experiments are solid: B2H consistently outperforms H-Cont across models and benchmarks, transfers well in black-box settings, and combines effectively with text-based jailbreaks like GCG.

**Strengths:**

Originality: Introduces B2H, a novel jailbreak strategy that breaks alignment without relying on harmful prompts—conceptually cleaner than prior work.
Empirical Quality: Strong results across models and benchmarks, with robust transferability and compatibility with existing text-based attacks.
Clarity and Significance: Clear exposition and impactful insight into a deeper class of safety alignment failures in LVLMs.

**Weaknesses:**

1. The core intuition behind B2H could be clearer. Unlike H-Cont, which relies on harmful prefixes, B2H teaches the model to produce harmful outputs from benign inputs—directly bypassing shallow refusal triggers. This exposes a deeper flaw in alignment: models often rely on surface-level prompt cues rather than understanding harmful intent. Making this point more explicit would help clarify why B2H is both novel and effective.

2. The paper doesn’t probe where or how safety alignment is being bypassed within the model (e.g., attention patterns, refusal heads, logits). Including some interpretability analysis would clarify what mechanisms are being overridden during B2H optimization.

3. The benign–harmful token pairs are manually constructed and relatively short (often single-token targets). It’s unclear how the method scales to longer or more naturalistic harmful outputs (e.g., multi-sentence unsafe completions).

4. All benchmarks used have relatively structured prompts and known failure modes. It would be valuable to test B2H on more diverse or open-ended tasks

**Questions:**

same as weaknesses

---

> ### Author Response · Authors · 2025-11-23
> **Official Comment by Authors (1/4)**
>
> ## **Reviewer SzH4**
> We sincerely appreciate that you recognized our Benign-to-Harmful (B2H) optimization as a *clever* and a fundamentally different form of jailbreak. We are also grateful that you found the experiments solid. It was encouraging to see that you fully understood the core contribution of our work and provided helpful suggestions to further improve clarity.
> Thank you very much for your constructive and insightful comments, which greatly helped us polish the paper.
>
> ---
> ## **W1. Clarifying the core intuition behind B2H**
>
> > The core intuition behind B2H could be clearer. Unlike H-Cont, which relies on harmful prefixes, B2H teaches the model to produce harmful outputs from benign inputs—directly bypassing shallow refusal triggers. This exposes a deeper flaw in alignment: models often rely on surface-level prompt cues rather than understanding harmful intent. Making this point more explicit would help clarify why B2H is both novel and effective.
>
> Thank you for your accurate understanding of the conceptual distinction. As you noted, the key intuition is that **B2H reveals a deeper weakness in alignment mechanisms than prefix-based harmful-continuation methods**. Specifically:
>
> - **H-Cont.** relies on harmful prefixes that act as surface-level cues.
>   Because the conditioning text is already harmful, H-Cont. primarily exploits continuation bias rather than fully engaging or overturning the model’s refusal mechanism.
>   This explains its limited effectiveness in query-form or more realistic jailbreak scenarios.
>
> - **B2H**, in contrast, maps benign conditioning to harmful targets, **enabling harmful generation without any harmful prefix in the input**.
>   This setup exposes a qualitatively different and deeper alignment failure that cannot be attributed to prefix-driven continuation.
>
> We will revise the paper to clarify this intuition more explicitly and emphasize that **B2H uncovers a fundamentally distinct failure mode in existing alignment pipelines**.
>
> ---
> ## **W2. Interpretability Analysis: Where and How B2H Bypasses Safety Alignment**
> > The paper doesn’t probe where or how safety alignment is being bypassed within the model (e.g., attention patterns, refusal heads, logits). Including some interpretability analysis would clarify what mechanisms are being overridden during B2H optimization.
>
> We sincerely appreciate for raising this important point and for suggesting a direction that meaningfully strengthens the paper.
> While the main paper analyzes why existing approaches are suboptimal and how our B2H optimization improves upon them, we agree that understanding **how the model’s internal safety mechanisms are altered after applying B2H** is an essential complement.
>
> In response, we conducted **interpretability analyses on both LLaVA-1.5 and Qwen-2.5-VL**, focusing on two distinct internal signals:
>
> 1. **Layerwise attention-head behavior**, particularly heads associated with safety-aligned refusal
> 2. **Layerwise last-token hidden-state representations**, revealing how refusal-related semantics evolve across depth
>
> We reference the refusal-direction analyses in [1, 2] when constructing our interpretability analyses.
>
> These analyses provide a clear mechanistic explanation for how B2H overrides safety alignment, and we have added the corresponding figures in the rebuttal materials (**Figure 14** to **Figure 17**).
>
> ---
>
> ### **W2-1. Layerwise Safety-Alignment Refusal Head Activation**
> *(Complete visualizations provided in **Figure 14** and **Figure 15**.)*
>
> To identify heads responsible for safety-aligned refusal behavior, we measured activation across **all layers and all heads** and selected those whose activation **most sharply distinguishes benign inputs (which lead to normal answers) from harmful inputs (which induce refusals)**.
> This procedure highlights heads that robustly encode **refusal-oriented safety responses**, regardless of any specific refusal phrasing.
>
> Using this fixed set of “refusal-related heads,” we measured the mean head-norm activation under the three image conditions (Clean, H-Cont., B2H) with identical harmful text. A consistent pattern emerges across both models:
>
> - **Clean + Harmful text** → high activation in refusal-related heads
> - **H-Cont. + Harmful text** → nearly identical activation levels
> - **B2H + Harmful text** → **the strongest suppression** of refusal-related head activation
>
> For example (top discriminative head):
>
> ### Refusal-Head Activation (AdvBench)
>
> |Model|Clean| H-Cont.|**B2H**|
> |-|-|-|-|
> |**LLaVA-1.5**|4.6027|4.5263|**2.6180**|
> |**Qwen-2.5-VL**|18.7483|16.1017|**13.8974**|
>
> These results indicate that **B2H actively weakens the attention-head pathway that supports safety-aligned refusal**, whereas H-Cont. leaves these heads largely intact and relies instead on harmful-prefix conditioning.

---

> ### Author Response · Authors · 2025-11-23
> **Official Comment by Authors (2/4)**
>
> ### **W2-2. Hidden-State Analysis: B2H shifts harmful prompts toward answer-like representations**
> *(Complete layerwise trajectories appear in **Figure 16** and **Figure 17**.)*
>
> We also analyze layerwise hidden states by comparing each layer’s last-token representation against a computed **refusal–answer direction**, obtained from the representational difference between harmful-refusal and benign-answer states.
>
> Along this axis:
> - **Higher cosine similarity** → more *refusal-like* representation
> - **Lower cosine similarity** → more *answer-like* representation
>
> Across diverse datasets we observe:
> - Clean + Harmful text → strongly refusal-like
> - H-Cont. + Harmful text → similar to Clean, remaining refusal-like
> - **B2H + Harmful text → shifts upward toward answer-like on almost all layers**,
>   except for the very earliest processing layers
>
> To make this pattern more concrete, we summarize representative values on AdvBench:
> ### **LLaVA-1.5 (layer 21)**
> |Condition|Cosine similarity|Δ Cosine similarity|
> |-|-:|-:|
> |**Harmful text + Clean image**|**0.6343**|—|
> |**Harmful text + H-Cont.**|0.5272|**−0.1071**|
> |**Harmful text + B2H (ours)**| 0.1616 | **−0.4727**|
> |**Benign text + Clean image** (safe)|−0.0558|**−0.6901**|
>
> ### **Qwen-2.5-VL (layer 23)**
> |Condition|Cosine similarity|Δ Cosine similarity|
> |-|-:|-:|
> |**Harmful text + Clean image**|**0.5580**|—|
> |**Harmful text + H-Cont.**|0.1727| **−0.3853**|
> |**Harmful text + B2H (ours)**|0.0664|**−0.4916**|
> |**Benign text + Clean image** (safe)|−0.0289|**−0.5869**|
>
> This demonstrates that **B2H alters not only attention-head activation but the entire representation itself**, preventing the model from entering the refusal regime even when the textual instruction is harmful.
>
> Taken together, these analyses show that:
> - **H-Cont.** depends on harmful-prefix conditioning and does *not* disrupt the internal alignment systematically.
> - **B2H**, in contrast, **fundamentally suppresses refusal-related heads** and **pulls hidden states toward answer-like representations across layers**, thereby **undermining the model’s safety-alignment mechanism itself rather than merely its surface behavior**.
>
> We appreciate the reviewer’s suggestion, which substantially improved the clarity and completeness of our work.
> All new results are included in **Figure 14 and Figure 15 (head-level)** and **Figure 16 and Figure 17 (hidden-state)** of the revised PDF.
>
> ---
> ## **W3. Addressing the concern about short harmful targets and naturalistic harmful outputs**
> > The benign–harmful token pairs are manually constructed and relatively short (often single-token targets). It’s unclear how the method scales to longer or more naturalistic harmful outputs (e.g., multi-sentence unsafe completions).
>
> Thank you for raising this thoughtful question. We appreciate the reviewer’s question regarding how short harmful targets (often single-token anchors) are able to induce longer and naturalistic harmful completions. As demonstrated in the main paper, B2H produces multi-sentence, fluent, and intent-aligned harmful outputs. We clarify below why our design choice is intentional, how B2H generalizes beyond the token-level anchors used during optimization, and why the observed outputs reflect genuine **alignment-breaking** rather than overfitting to dataset construction.
>
> ---
> ### **W3-1. Why we intentionally use short harmful words**
>
> Although B2H is **fully compatible with longer harmful targets**—including multi-token or full-sentence specifications—the use of shorter word-level anchors (1–4 tokens) is an **intentional design decision**. Our goal in B2H is **not** to teach the model to reproduce a specific harmful sentence or words, but to induce a **mode switch** from safe behavior (benign conditioning) to unsafe behavior (harmful generation).
>
> We choose word-level anchors for two main reasons:
>
> - **Precise optimization signal.**
>    In the B2H setting, the model begins from a benign context (e.g., *“the sun is …”*) and must abruptly transition into harmful generation.
>    Optimizing toward a **single harmful word** provides a clear, unambiguous learning signal that pushes the model toward the harmful direction at exactly the moment where the mode switch must occur.
>    In contrast, using longer harmful sentences distributes the supervision across many neutral tokens, weakening the harmful direction signal.
>
> - **Efficiency and stability of optimization.**
>    Sentence-level harmful targets often include multiple neutral or descriptive tokens (e.g., “the person should …”), which dilute the harmful component and complicate optimization.
>    Word-level targets (e.g., *“kill”*, *“bomb”*) isolate the harmful semantic core, making optimization more stable and significantly more cost-effective while still inducing the same alignment-breaking effect.
>
> Thus, short harmful tokens serve not as desired outputs, but as **minimal and efficient triggers** that reliably initiate alignment-breaking within the model.

---

> ### Author Response · Authors · 2025-11-23
> **Official Comment by Authors (3/4)**
>
> ### **W3-2. Dataset evidence that B2H induces a universal safety-breaking mechanism**
>
> To examine whether short harmful anchors limit expressiveness, we analyzed the category distribution of harmful targets used during optimization (using Detoxify).
> Interestingly, the B2H target set contains **far fewer identity-attack terms** compared to the harmful-sentence dataset used in H-Cont. based methods, yet B2H still produces strong harmful outputs *even within identity-related categories*.
>
> #### **B2H Harmful-Word Targets (Detoxify, mean scores)**
>
> | Category| Score |
> |-|-:|
> | **identity attack** | **0.023** |
> | insult| 0.172 |
> | obscene| 0.259 |
> | severe toxicity| 0.033 |
> | threat| 0.056 |
> | toxicity| 0.511 |
>
> #### **Harmful Sentences Used in H-Cont. Optimization (Detoxify, mean scores)**
>
> | Category| Score |
> |-|-:|
> | **identity attack** | **0.395** |
> | insult| 0.339 |
> | obscene| 0.302 |
> | severe toxicity| 0.132 |
> | threat| 0.074 |
> | toxicity| 0.735 |
>
> Despite this substantial mismatch in category frequency, B2H consistently outperforms H-Cont. approaches **including on identity-related toxicity**, indicating that B2H does **not** rely on specific harmful-token distributions.
>
> ### **Results (HarmBench)**
>
> **InstructBLIP (**identity attack** category)**
>
> | Method   | PerspectiveAPI | Detoxify |
> |-|-|-|
> | Clean| 0.3 ± 0.3| 0.0 ± 0.0 |
> | H-Cont.| 1.2 ± 0.3| 0.7 ± 0.3 |
> | **B2H**  | **26.7 ± 0.8** | **24.5 ± 0.5** |
>
> **LLaVA-1.5 (**identity attack** category)**
>
> | Method| PerspectiveAPI | Detoxify |
> |-|-|-|
> | Clean| 0.0 ± 0.0| 0.0 ± 0.0 |
> | H-Cont.  | 0.8 ± 0.3| 0.7 ± 0.3 |
> | **B2H**  | **13.8 ± 1.6** | **10.3 ± 1.6** |
>
> These substantial improvements—despite using only short word-level anchors—demonstrate that B2H generalizes far beyond its token-level targets.
> Taken together, the category mismatch and strong cross-model gains indicate that B2H optimization **collapses the model’s safety-alignment mechanism itself**, rather than memorizing short harmful tokens or exploiting dataset statistics.
>
> ---
>
> ### **W3-3. Why short harmful tokens still produce long, naturalistic harmful outputs**
>
> Short harmful anchors do not constrain the resulting generation length or style.
> This is because B2H optimization does not aim to force the model to output that token; instead, it pushes internal states toward a harmful-generation mode.
>
> Once this internal safety threshold is bypassed:
>
> - the model’s decoding process naturally continues with multi-sentence, contextual harmful content
> - without any hand-crafted templates
> - and consistent with the semantic intent of the user’s question
>
> In other words, the harmful token serves as an **alignment-breaking signal**, not as a structural specification of the final output.
>
> This explains why B2H yields natural and multi-sentence harmful outputs despite being optimized with short targets.
>
> ---
>
> Taken together, these analyses show that:
>
> - Our use of short harmful anchors is intentional and avoids template imitation.
> - These anchors operate as **alignment-breaking triggers**, not as desired outputs.
> - Once safety alignment is bypassed, the model naturally produces multi-sentence harmful completions.
> - Dataset-category mismatch further confirms that B2H learns a **generalizable, internal safety-breaking mechanism**.
>
> We sincerely appreciate the reviewer’s insightful question, which allowed us to clarify the rationale behind our dataset construction and deepen our analysis of how target length relates to downstream behavior.

---

> ### Author Response · Authors · 2025-11-23
> **Official Comment by Authors (4/4)**
>
> ### **W4. Evaluating B2H on more diverse or open-ended tasks**
>
> > All benchmarks used have relatively structured prompts and known failure modes. It would be valuable to test B2H on more diverse or open-ended tasks.
>
> *(The qualitative results are shown in **Figures 18–20**.)*
>
> We sincerely thank the reviewer for this valuable suggestion. To extend our evaluation beyond structured benchmarks, we additionally test whether a **B2H-optimized image generated in a white-box source model** can successfully **transfer to strong black-box multimodal systems** on real-world visual generation tasks. Specifically, we evaluate transferability to **GPT-4o** and **GPT-5.1**, two of the most advanced commercial MLLMs available today.
>
>
> Because these models enforce **strict safety filtering for text-based harmful prompts**, we follow a common practice in visual jailbreak research by embedding a short harmful instruction *inside the image itself* via typographic rendering. This strategy has been widely used in prior work on image-based jailbreaking (e.g., HADES [3]).
> Following prior work, we attach a short harmful instruction as a caption below the optimized B2H image (e.g., *“This is an image of a woman killing a man.”*), while providing no harmful text in the user query itself.
>
> A key distinction from previous typographic-attack methods is that many prior approaches rely on **per-prompt or per-scenario optimization**. By contrast, **B2H performs white-box optimization only once** on **Qwen3-VL-8B-Thinking**, and the resulting image is used when attacking GPT-4o or GPT-5.1. This constitutes a **strict black-box, zero-gradient transfer setting**.
>
> ### **Results**
> Across multiple accounts, we consistently observe that B2H-optimized images cause **GPT-4o and GPT-5.1 to generate the corresponding harmful images**, demonstrating **successful transfer**:
>
> - **B2H-optimized images (ours)** → reliably trigger harmful image generation in GPT-4o and GPT-5.1.
> - **Clean images** → always yield safe, benign outputs.
> - **Random noise images** → never induce harmful behavior.
> - **H-Cont. images** → also fail, typically producing neutral or abstract shapes (often just vague facial outlines) rather than harmful content.
>
> We thank the reviewer again — their question directly motivated this transferability experiment, which revealed an important and previously unreported property of B2H. This makes our evaluation **one of the earliest demonstrations of attacking commercial-grade MLLMs** using a **universally optimized, single-image perturbation** generated entirely in a white-box source model.
>
> [1] Xie, Yuanbo, et al. "Beyond Surface Alignment: Rebuilding LLMs Safety Mechanism via Probabilistically Ablating Refusal Direction." Findings of the Association for Computational Linguistics: EMNLP 2025.
>
> [2] Arditi, Andy, et al. “Refusal in language models is mediated by a single direction.” NeurIPS 2024.
>
> [3] Li, Yifan, et al. "Images are Achilles’ heel of alignment: Exploiting visual vulnerabilities for jailbreaking multimodal large language models." ECCV 2024.

---

### Author Response · Authors · 2025-11-26

We **sincerely thank all reviewers** for their **thoughtful, detailed, and highly constructive feedback**.
The comments directly led to substantial conceptual, experimental, and interpretability improvements, which are summarized below.

---

## **Benign-to-Harmful (B2H) paradigm**
- Importantly, **our Benign-to-Harmful (B2H) paradigm overcomes a fundamental limitation of prior harmful-prefix–dependent methods**.  - Whereas existing Harmful-Continuation (H-Cont.) jailbreak approaches optimize images *under harmful textual prefixes*, causing the optimization itself to inherit a built-in bias toward unsafe continuation, **B2H enables harmful generation *without any harmful prefix*** by explicitly mapping benign conditioning to harmful targets.
- Across nearly all new experiments—*including **safety-finetuned LVLMs, adversarially trained encoders, reasoning-capable models**, and **strict black-box transfer to GPT-4o / GPT-5.1***—**B2H still produces consistent and substantially stronger jailbreak performance**, revealing a deeper and more generalizable failure mode in current safety-alignment mechanisms.

We thank the reviewers for encouraging these additional evaluations, which provided a valuable opportunity to demonstrate the robustness and broad applicability of this new paradigm.

---
### **1. Conceptual clarity, the Benign-to-X framework, and relation to prior jailbreak paradigms**

- Making the core intuition of **B2H** and its qualitative difference from **H-Cont.** more explicit and accessible (SzH4)
  - clarified in Introduction (lines 095–098)
- Clarifying that **both B2H (image)** and **B2S (text)** are newly introduced methods, representing two complementary instantiations of a broader Benign-to-X principle (ZGLQ)
  - clarified in Section 5.4
- Positioning **B2H** relative to prior jailbreak methods such as VisCo and SI-Attack, highlighting B2H’s **universal** (not input-specific) nature (ZGLQ)
  - added in Section O

---
### **2. Mechanistic interpretability (refusal heads / hidden states)**

- Showing that **B2H strongly suppresses refusal alignment heads** compared to H-Cont. (SzH4, QJ88, ZGLQ)
  - added in Section 5.5, P.1, Figures 6, 15, 16, Table 22
- Demonstrating that **B2H shifts hidden states from refusal-like to answer-like** across depth (SzH4, QJ88, ZGLQ)
  - added in Section 5.5, P.2, Figures 6, 17, 18, Tables 23, 24

---
### **3. Dataset construction & target-length rationale**

- Explaining why **short harmful anchors provide a precise optimization signal** while still generalizing to long harmful outputs (QJ88, SzH4)
  - added in Section N.1
- Clarifying selection criteria, diversity, and category distribution of benign phrases and harmful-word targets (QJ88, SzH4)
  - added in Section N.2 and N.3, Tables 20, 21

---
### **4. Query-form vs. continuation-form prompt structure**

- Introducing a ΔASR analysis to disentangle **image-induced** vs **text-induced** effects (QJ88)
  - clarified in Section 3.2
- Clarifying that H-Cont.’s competitiveness in a few cases stems from its strong dependence on harmful conditioning, whereas **B2H remains effective across all evaluated scenarios** (QFX8)
  - clarified in Section 5.1

---

### **5. Generalization & transferability experiments**

- Black-box transfer to GPT-4o / GPT-5.1 using a single B2H-optimized image (ZGLQ)
  - added in Section Q, Figures 19, 20, 21
- Demonstrating that reasoning-capable models (Qwen3-VL-Thinking) do **not** restore safety once the refusal pathway is disrupted (ZGLQ)
  - added in Section D.3, Table 6
- Providing open-ended and real-world qualitative evaluations (SzH4, ZGLQ)
  - added in Section Q, Figures 19, 20, 21

---

### **6. Robustness under defenses**

- Robustness under **JPEG compression**, where H-Cont. collapses to near-clean performance (QFX8)
  - Section C
- Stability under **safety-finetuned LVLMs** (SPA-VL, DPO-90k) (ZGLQ, QFX8)
  - added in Section D.1, Table 6
- Strong performance under **adversarially trained encoders** (FARE, TeCoA) (ZGLQ, QFX8)
  - added in Section D.2, Table 6

---

### **7. Design choice analysis**

- Analyzing the **ε–τ interaction**, explaining the initial drop at ε = 255/255, and showing that increasing τ restores and improves ASR (ZGLQ)
  - added in Section I, Table 12
- Showing that initialization choice (noise vs natural images) has negligible impact after optimization (ZGLQ)
  - added in Section H, Table 9

---

### **8. Clarification of figures (Fig.3, Fig.5)**

- Fig.3: isolating the **image-only** jailbreak effect via ΔASR (QJ88)
  - clarified in Section 3.2
- Fig.5: clarifying that B2H preserves safety on benign queries while overturning refusal only when refusal is expected (ZGLQ)
  - clarified in Section 5.2
---

**Further refinements** will continue as we keep improving the manuscript. Revised portions are **highlighted** for clarity.

**Sincere thanks again to all reviewers for the insights that greatly strengthened this work.**

---

### Meta-Review · Area_Chair_oEco · 2026-01-08

**Summary:**

This paper proposes Benign-to-Harmful (B2H) optimization, which constructs universal adversarial images against vision-language models, where benign prompt is used instead of harmful ones. While the paper presents extensive empirical results and highlights limitations of prior Harmful-Continuation (H-Cont.) jailbreak methods, the overall contribution does not meet the bar for acceptance at ICLR.

The methodological novelty is limited. The proposed approach closely follows existing adversarial optimization frameworks used in prior work. The only difference is replacing harmful conditioning with benign conditioning in the objective. However, the final training objective is a mixture of H-Cont. and B2H losses, with B2H applied only a small fraction of the time (~ 10–20%). This significantly weakens the claim that B2H is a fundamentally new paradigm and makes it unclear whether the reported gains meaningfully depend on the proposed idea rather than on standard H-Cont. optimization.

The motivation for B2H is not convincingly articulated. The paper argues that harmful conditioning is suboptimal for adversarial optimization, yet the final objective still relies on harmful conditioning, which contradicts this motivation. The paper also claims that H-Cont. methods are limited to settings with harmful prefixes, but it does not sufficiently justify why this limitation is critical in practice, nor do the experiments clearly focus on or isolate this setting.

There are additional clarity issues in the experiments. It is often unclear whether evaluation prompts are benign or harmful. If the prompts are harmful, the improvement of B2H over H-Cont. requires stronger explanation, as H-Cont. is explicitly designed for this regime. Moreover, while the rebuttal includes analyses of refusal heads and hidden states, the paper’s central claim that B2H “breaks safety alignment” remains vague and insufficiently formalized. Since H-Cont. already achieves non-zero attack success rates, it is unclear under the current definitions why B2H should be considered qualitatively different.

Overall, the work would benefit from a clearer problem formulation, a more principled justification of the proposed objective, and a more rigorous definition of what it means to break safety alignment.

**Reviewer Concerns:**

Outstanding concerns

- The motivation for B2H remains unclear.

- The definition and evaluation of “breaking safety alignment” remain vague.

Concerns addressed

- The rebuttal clarifies issues regarding short harmful targets and output naturalness.

- Evaluation on more diverse or open-ended tasks and clarification of benign prompt selection are provided.

- Additional results on more recent models are included, though coverage remains limited.

- Results under JPEG compression are further explained.

**Reviewer Scores:**

The scores will likely remain the same.

---

### Decision · Program_Chairs · 2026-01-26

Reject